# Investigating the Potential of Transmucosal Delivery of Febuxostat from Oral Lyophilized Tablets Loaded with a Self-Nanoemulsifying Delivery System

**DOI:** 10.3390/pharmaceutics12060534

**Published:** 2020-06-10

**Authors:** Yasir A. Al-Amodi, Khaled M Hosny, Waleed S. Alharbi, Martin K. Safo, Khalid M El-Say

**Affiliations:** 1Department of Pharmaceutics, Faculty of Pharmacy, King Abdulaziz University, Jeddah 21589, Saudi Arabia or yalamodi0003@stu.kau.edu.sa (Y.A.A.-A.); Kmhomar@kau.edu.sa (K.M.H.); wsmalharbi@kau.edu.sa (W.S.A.); 2Department of Pharmaceutics and Industrial Pharmacy, Faculty of Pharmacy, Beni-Suef University, Beni-Suef 62511, Egypt; 3Department of Medicinal Chemistry, and the Institute for Structural Biology, Drug Discovery and Development, School of Pharmacy, Virginia Commonwealth University, Richmond, VA 23298, USA; msafo@vcu.edu; 4Department of Pharmaceutics and Industrial Pharmacy, Faculty of Pharmacy, Al-Azhar University, Cairo 11651, Egypt

**Keywords:** bioavailability, febuxostat, gout, lyophilized tablets, self-nanoemulsifying delivery system

## Abstract

Gout is the most familiar inflammatory arthritis condition caused by the elevation of uric acid in the bloodstream. Febuxostat (FBX) is the latest drug approved by the United States Food and Drug Administration (US FDA) for the treatment of gout and hyperuricemia. FBX is characterized by low solubility resulting in poor gastrointestinal bioavailability. This study aimed at improving the oral bioavailability of FBX by its incorporation into self-nanoemulsifying delivery systems (SNEDS) with minimum globule size and maximum stability index. The SNEDS-incorporated FBX was loaded into a carrier substrate with a large surface area and lyophilized with other excipients to produce a fluffy, porous-like structure tablet for the transmucosal delivery of FBX. The solubility of FBX was studied in various oils, surfactants, and cosurfactants. Extreme vertices design was utilized to optimize FBX-SNEDS, and subsequently loaded into lyophilized tablets along with suitable excipients. The percentages of the main tablet excipients were optimized using a Box–Behnken design to develop self-nanoemulsifying lyophilized tablets (SNELTs) with minimum disintegration time and maximum drug release. The pharmacokinetics parameters of the optimized FBX-SNELTs were tested in healthy human volunteers in comparison with the marketed FBX tablets. The results revealed that the optimized FBX-SNELTs increased the maximum plasma concentration (C_max_) and decreased the time to reach C_max_ (T_max_) with a large area under the curve (AUC) as a result of the enhanced relative oral bioavailability of 146.4%. The significant enhancement of FBX bioavailability is expected to lead to reduced side effects and frequency of administration during the treatment of gout.

## 1. Introduction

Hyperuricemia is a metabolic disorder that is characterized by an increase in serum urate over a threshold value that leads to the growth of crystals of monosodium urate in tissues and around the joints, and when it becomes symptomatic in a patient, it is diagnosed as gout [1]. Gout is the most familiar inflammatory arthritis condition in men above 40 years, although it is increasingly occurring in postmenopausal women [2]. In addition, the prevalence of gout has kept pace with increasing population growth. For example, in China 15.3 million were diagnosed with chronic gout in 2013, and the number is expected to increase to 17.7 million in 2021 [3].

Febuxostat (FBX) is the latest drug developed for hyperuricemia and gout treatment [4]. The United States Food and Drug Administration (US FDA) and European Medicines Agency (EMA) approved FBX for patients who have an inadequate reduction in serum uric acid level or who do not tolerate allopurinol [5]. At 80 mg and 120 mg doses, FBX was more potent at decreasing serum uric acid level than allopurinol at doses of 200 mg and 300 mg as demonstrated by clinical comparative studies [6]. Of note is that FBX, which is practically insoluble in water [7], offers dissolution rate limited-absorption, consequently leading to poor and variable oral bioavailability [8].

Owing to its massive applicative potential, nanotechnology is considered one of the foremost technologies of the 21st century. In the pharmaceutical field, nanotechnology has led to improvement in the efficacy both of novel and old drugs, with the potential to offer state-of-the-art solutions for all therapeutic agents, and diagnostic tools for varied diseases [9,10].

Self-nanoemulsifying delivery systems (SNEDS) and the nanometric-sized transparent systems produced on dilution, known as nanoemulsions (NEs), have been widely studied for their abundant potential applications [11,12,13,14,15,16]. SNEDS offered greater stability when compared to other lipid-based drug delivery systems [17,18]. Also, SNEDS improved the solubility and subsequently the oral bioavailability of lipophilic drugs [19].

Due to potential incompatibility and liquid leakage from capsules filled with liquid SNEDS, as well as expensive capsule liquid-filling technology, solid SNEDS are desired [20]. Solid SNEDS can be loaded onto a solid carrier to provide solid dosage forms to augment drug stability, dose accuracy, and patient convenience [21,22,23]. Porous carriers, such as silicon dioxide, magnesium aluminometasilicate, dibasic calcium phosphate, and calcium silicate have been used in the solidification of SNEDS for their great potential in adsorbing liquid [24,25,26,27,28]. The lyophilization technique brings numerous advantages of freeze-dried dosage forms such as quick disintegration, excellent stability, and respectable preservation. In this process, water is sublimated directly from the solid ice state into the vapor gas state. This will permit less damage to the substance than any other drying technique that depends on a higher temperature. Furthermore, flavors, odors, and other excipients are generally unaffected. Also, the lyophilization process creates a porous-like structure that promotes the rapid rehydration and solubilization of the tablet. The produced lyophilized tablet, when placed in the mouth, can be rapidly dispersed or dissolved in saliva without the need for water or chewing and can be swallowed in the form of a liquid, increasing its dissolution and absorption, and consequently its bioavailability [23,29]. Therefore, the development of self-nanoemulsifying lyophilized tablets (SNELTs) is highly preferred owing to their scalability and robustness, and their ability to gain all the advantages of a liquid system. A variety of sugars have been used as cryoprotectants during the dehydration of the tablets [30].

The aim of this study was to improve the poor bioavailability of FBX by multiconcerted mechanisms, primarily by utilizing nanotechnology through the preparation of FBX in the form of SNEDS, utilizing mixture design to minimize the globule size into a nanosized range to improve FBX solubility. The FBX-SNEDS were loaded into fumed silica to provide a large surface area and improve their dispersion into an aqueous gelatin solution. Then, the other excipients such as Croscarmellose sodium, lactose, and xylitol were subsequently added. All the components were homogeneously mixed until the formation of a slurry that was poured into empty pockets of tablet blister packs that were finally lyophilized. The incorporation of the resultant mixture in a lyophilized tablet produced a tablet with a fluffy, porous-like structure, which was then optimized using the Box–Behnken design (BBD). The optimized tablet was subjected to an in vivo pharmacokinetic study on human volunteers in comparison with the marketed tablets. The resulting oral transmucosal FBX-SNELTs afford a porous tablet with fast dissolution and high bioavailability and are expected to lead to better patient compliance.

## 2. Materials and Methods

### 2.1. Materials

Febuxostat (FBX) powder was a kind gift from SPIMACO Addwaeih’s. (Riyadh, Saudi Arabia). Propylene glycol and clove oil were purchased from TEDIA Company, Inc. (Fairfield, OH, USA). Olive oil, linoleic acid 60% and isopropyl myristate 96% were purchased from Acros Organics (Morris Plains, NJ, USA). Microcrystalline cellulose (Avicel^®^ PH-101), fumed silica (0.007 µm), gelatin, Tween 80, Tween 20, polyethylene glycol (PEG) 200 and 400, triacetin, oleic acid, Lauroyl polyoxylglycerides (Gelucire^®^ 44/14, hydrogenated coconut oil with PEG 32), mannitol, castor oil, methanol, polyethylene glycol 40 stearate (Myrj^®^ 52), glycerin, hydroxypropyl methylcellulose (HPMC), xylitol, croscarmellose sodium (Ac-Di-Sol^®^), and starch were purchased from Sigma-Aldrich (St. Louis, MO, USA). Propylene glycol dicarprylate, Labrafil^®^ M1944, Labrafac^®^ WL1349, Labrasol^®^, Transcutol^®^, and Kolliphor^®^ were purchased from Gattefosse (Saint-Priest, France). Lactose was obtained from the Spectrum chemical manufacturing corporation (Gardena, CA, USA). All other chemicals and solvents were of analytical grades.

### 2.2. Solubility Studies of FBX in Different Self-Nanoemulsion Components

FBX solubility in oils, namely Labrafac^®^ WL1349, castor oil, olive oil, clove oil, oleic acid, isopropyl myristate 96%, Lauroyl polyoxylglycerides, triacetin, and linoleic acid 60%, was determined. Also, FBX solubility was determined in surfactants such as Tween 80, Tween 20, polyethylene glycol 40 stearate, propylene glycol dicarprylate, Labrafil^®^ M1944, glyceryl distearate, Labrasol^®^, and Kolliphor^®^. Solubility in cosurfactants like PEG 200, PEG 400, propylene glycol, Transcutol^®^, and glycerin was also determined. The experiment was performed by dissolving an excess amount of FBX in 3 mL of each liquid separately. The mixture was shaken in a thermostatically controlled shaking water bath (Model 1031; GFL Corporation, Burgwedel, Germany) at 25 ± 0.5 °C for 48 h till equilibrium. 1ml of this mixture was centrifuged at 3000 rpm for 20 min and the concentration of FBX was determined spectrophotometrically at 316 nm using a UV-VIS spectrophotometer (Jenway 7315; Bibby Scientific Limited, Stone, UK).

### 2.3. Construction of Pseudo-Ternary Phase Diagram

Based on the solubility studies, a ternary phase diagram was established using the chosen oil (castor oil), surfactant (polyethylene glycol-40-stearate), and cosurfactant (Transcutol^®^) to identify the levels of self-nanoemulsion component, which spontaneously form a clear NE after dilution with water.

### 2.4. Formulation of FBX-Loaded SNEDS according to the Mixture Design

The extreme vertices design of the special cubic model was utilized to statistically optimize the effects of SNEDS components in a randomized order [31]. The three-component system was planned to use the percent of oil phase (castor oil (X_1_)), the percent of surfactant (polyethylene glycol 40 stearate (X_2_)), and the percent of cosurfactant (Transcutol^®^ (X_3_)) to develop a SNEDS with minimum globule size and maximum stability index. The dependent variables were the mean globule size (Y_1_) and the stability index (Y_2_). The components and their ratios selected to perform the mixture design are summarized in Table 1. For any mixture, the total of the three components always added to 100%. The correlations between the components and the obtained responses were statistically analyzed using the statistical package Statgraphics^®^ Centurion 18 Software (StatPoint, Inc., Warrenton, VA, USA).

### 2.5. Evaluation of the FBX-NE Formulations

#### 2.5.1. Visual Inspection for Emulsification Ability

The FBX-SNEDS was inspected visually for its clarity and its ability to be emulsified spontaneously upon mixing of its components. Briefly, a specific weight (50 mg) of the SNEDS formulation was placed into 100 mL of distilled water and observed visually for the emulsification ability. The appearance of the NE after gentle agitation was graded as very cloudy, cloudy, or clear. Visual observations were made immediately after dilution for spontaneous emulsification, transparency, phase separation, and drug precipitation [16].

#### 2.5.2. Globule Size Determination

Aliquots of 20 mL distilled water containing 100 mg of each formulation were used to determine the globule size by dynamic light scattering using a Zetatrac particle size analyzer from Microtrac Inc. (Montgomeryville, PA, USA).

#### 2.5.3. Thermodynamic Stability Studies

The NE formulations were examined for their thermodynamic stability [32]. The formulations were exposed to three freeze-thaw cycles, with each cycle comprising 12 h freezing at −20 °C, followed by 12h thawing at +25 °C. The resultant formulations were examined for a change in globule size to ensure the stability of the NE. The stability index for the NE was calculated from Equation (1).
Stability index = [(Initial size – Change in size)/Initial size] × 100(1)

#### 2.5.4. Morphology of NE

A transmission electron microscope (TEM, H7500, Hitachi, Japan) was used to analyze the morphology and structure of the optimized formulation. The formulation was further used to prepare the FBX-SNELTs.

### 2.6. Preparation of FBX-SNELTs

FBX-SNELTs were prepared according to a previously reported method [23]. Briefly, the specified weight of FBX-SNEDS was mixed with 200 mg fumed silica, 100 mg xylitol, 100 mg mannitol, and 200 mg lactose. Croscarmellose sodium percentage, gelatin solution concentration, and hydroxypropyl methylcellulose percentage were added in varying percentages to study their effect on the in vitro disintegration and in vitro dissolution of the prepared FBX-SNELTs.

### 2.7. Optimization of FBX-SNELTs

The results of the fifteen formulations were evaluated using analysis of variance (ANOVA) followed by multiple response optimization with the Statgraphics software. The optimum concentrations for the three variables were established to develop FBX-SNELTs with minimum disintegration time and an optimum drug release profile. The optimized FBX-SNELT was prepared and evaluated for weight uniformity, thickness, content uniformity, in vitro disintegration, and in vitro dissolution.

### 2.8. In Vivo Pharmacokinetic Studies

An open-label, one-period, parallel design comprising two weeks of screening preceding 24 h study periods was conducted. The chosen volunteers (6 male and 6 female with ages ranging between 22 and 39) were given a single dose of either optimized FBX-SNELT equivalent to 20 mg (test), or marketed FBX tablet equivalent to 20 mg (reference). The study was approved by the Ethics Committee of the General Hospital of Beni-Suef University, Egypt with approval number FMBSU-106-19 in March 2019. Also, the study was conducted according to the Helsinki agreement protocol. An hour before dosing, a cannula was inserted into the volunteer’s forearm and kept there for 24 h. After the administration of the formulation, blood samples were collected at a predetermined time interval of 24 h. The plasma was separated and the concentration of FBX in each sample was determined using a Shimadzu Prominence HPLC system (Shimadzu Corp., Kyoto, Japan) with the fluorescence detection method at excitation and emission wavelengths of 320 and 380 nm, respectively [33]. In brief, after the addition of an internal standard (2-naphthoic acid), the proteins were removed from plasma samples (0.25 mL) by the addition of 0.25 mL of acetonitrile, mixed, and centrifuged, and the resulting supernatant was acidified with 25 µL of glacial acetic acid. FBX and the internal standard were resolved from the matrix components using a Nucleodur MN-C18 column, 5 µm, 250 × 4.6 mm (Macherey-Nagel, Düren, Germany). The mobile phase comprised 0.03% glacial acetic acid in water and acetonitrile (55:45, v/v). FBX showed a linear calibration curve ranging from 0.005 to 25µg mL^−1^ with a correlation coefficient of >0.997 and with a minimum limit of quantification with a 250 µL plasma sample of 0.005 µg mL^−1^. The pharmacokinetic parameters were determined by a non-compartmental pharmacokinetic model utilizing PK Solver 2.0 software (an add-in program for pharmacokinetic data). The results were expressed as mean ± SD and analyzed using GraphPad Prism 8 software (San Diego, CA, USA).

## 3. Results and Discussion

### 3.1. Solubility Studies

Figure 1 showed the solubility of FBX in the various oils, surfactants, and cosurfactants. The highest solubility of FBX in the oils was observed with castor oil (228.37 mg/mL) (Figure 1a); in the surfactants, with polyethylene glycol-40-stearate (435.2 mg/mL) (Figure 1b); and in the cosurfactants, with Transcutol^®^ (593.3 mg/mL) (Figure 1c). The surfactants Labrasol, Tween 20, Tween 80 and Kolliphor, as well as the cosurfactant PEG 400, also resulted in significant solubility of FBX (Figure 1).

The ability of a nanoemulsion (NE) to maintain a drug in solubilized form is markedly affected by the solubility of the drug in the oil phase [34]. Castor oil has been massively used in the pharmaceutical field as a solvent for poorly soluble drugs prepared in the form of an emulsion. It has been suggested that the presence of a hydroxyl functional group in ricinoleic acid, the main constituent of castor oil, aids the stability of the formed emulsion [35]. For surfactants, the nonionic polyethylene glycol 40 stearate is is commonly used in the formulation of NEs for oral or parenteral use due to its limited toxicity and high biocompatibility. Polyethylene glycol 40 stearate also has a suitable hydrophilic-lipophilic balance (HLB) value of 17 [36]. Surfactants with an HLB value greater than 10 are usually utilized in the formulation of emulsions intended to form a fine oil in water (o/w) NE when dispersed in gastrointestinal fluids [37]. Cosurfactants, e.g., Transcutol^®^, are usually used to obtain stable NE systems as they further decrease the interfacial tension between the oily and aqueous phases of the emulsion, and enhance the fluidity of the interface [34]. In the present study, castor oil, polyethylene glycol 40 stearate, and Transcutol^®^, which incidentally resulted in a high solubility of FBX, were selected as the components of choice for the formulation of FBX-SNEDS.

### 3.2. Construction of Pseudo-Ternary Phase Diagram

A pseudo-ternary phase diagram was established to identify the levels of SNEDS components which spontaneously form clear NEs after dilution with water as a preliminary test for the mixture design. Figure 2a shows the levels of the components that lie in the area in which the formed mixture spontaneously gives a NE after dilution. These levels were 10% to 15% (*w/w*) for oil, 40% to 60% (*w/w*) for surfactant, and 30% to 50% (*w/w*) for cosurfactant.

### 3.3. Optimization of FBX-NE Formulations

#### 3.3.1. Effect of NE Components on the Globule Size

The composition of the SNEDS formulations and the mean of the observed Y_1_ and Y_2_ are listed in Table 2. The results of the globule size varied from 175.7 nm to 452.8 nm and these results fitted to the special cubic model with a *p*-value of 0.0004. This variability in the globule size of the formulations resulted from the difference in the proportion of NE components. The contour plot displays the effect of the mixture components on the globule size of the NE as demonstrated in Figure 2b. The regression equation of the fitted special cubic model for the globule size (Y_1_) was generated using Equation (2).
Globule size (Y_1_) = 2043.42 X_1_ + 193.121 X_2_ + 278.409 X_3_ − 1627.05 X_1_ X_2_ − 1418.95 X_1_ X_3_ + 43.1492 X_2_ X_3_ + 468.679 X_1_ X_2_ X_3_.(2)

This equation and the two-dimensional contour plot demonstrated that a high proportion of surfactant (X_2_) and a low proportion of both oil (X_1_) and cosurfactant (X_3_) minimized the globule size of the formulation. The grey area in the system adjacent to the corner of the surfactant in the triangle signifies the minimum globule size of the formulations. The globule size was affected by the concentration of oil. The average globule size was found to be increased significantly in the formulations containing the highest level of oil (NE-3, NE-4, and NE-12). The globule size increased with increasing oil concentration, which has been suggested to be the main reason for the inadequate amount of surfactant/cosurfactant required to cover oil droplets and coalesce the globules [14,38]. Consistently, formulations with the lowest level of oil (NE-1, NE-2, NE-9, and NE-14) showed a smaller globule size, which can be attributed to the use of an appropriate concentration of surfactant/cosurfactant mixture. This provided a sufficient reduction in the free energy of the system, which afforded a strong mechanical barrier protecting the formed globules from coalescence. Systems with a mean globule size below 200 nm achieve the criteria for SNEDS [39].

#### 3.3.2. Effect of NE Components on the Stability Index

The results varied from 59% in NE-3 to 91% in NE-2, which were fitted to the special cubic model with a *p*-value of 0.0001. The contour plot shows the effect of the NE components on the stability index of the formulations as depicted in Figure 2c. The regression equation of the fitted special cubic model for the stability index (Y_2_) was generated using Equation (3).
Stability index (Y_2_) = −41.223 X_1_ + 63.548 X_2_ + 89.798 X_3_ + 120.573 X_1_ X_2_ + 143.673 X_1_ X_3_ − 6.346 X_2_ X_3_ − 37.568 X_1_ X_2_ X_3_.(3)

Equation (3) and the triangular two-dimensional contour plot demonstrated that the high proportion of cosurfactant (X_3_) and the low proportions of both oil (X_1_) and surfactant (X_2_) maximized the stability index of the formulations. The dark blue area in the system adjacent to the corner of the cosurfactant in the triangle represents the maximum stability index of the formulations. The results show that the stability index of formulations is affected by the concentration of cosurfactant. The formulation (NE-2) containing the highest level of cosurfactant showed the highest stability index. In contrast, formulations (NE-1, NE-3, NE-10, and NE-14) that contained low levels of cosurfactant showed the lowest stability index. The thermodynamic stability of NEs arises from their extremely low interfacial tension. In general, surfactants alone cannot decrease the interfacial free energy sufficiently, so the addition of cosurfactant is necessary to produce thermodynamically stable NE systems [40]. The same finding has been reported for microemulsion-based anthocyanin systems that showed that the stability of the system was improved by increasing the concentration of cosurfactant [41].

#### 3.3.3. Multiple Response Optimization Using the Desirability Function

The triangular dimensional contour plot displays the effect of the levels of the components on the desirability function after the optimization of NE as demonstrated in Figure 2d. It is obvious from the figure that the high proportions of cosurfactant (X_3_) and the low proportions of both oil (X_1_) and surfactant (X_2_) maximize the desirability of the formulation. The violet area in the system adjacent to the corner of the cosurfactant in the triangle represents the maximum desirability of the formulation. The composition of NE-2 as depicted in Table 2 resulted in minimum globule size and maximum stability index, and was therefore used in the preparation of FBX-SNELTs.

### 3.4. Morphological Examination Using TEM

Globule size is a critical factor in SNEDS performance because it influences the rate and extent of drug release, as well as drug absorption [39]. TEM images with different magnifications (Figure 3) revealed that the formed colloidal dispersion is characterized by uniform globule size distribution, a nano-size range of approximately 200 nm, and no globule aggregation. This globule size agreed with the results measured by Zetasizer as given in Table 2.

### 3.5. Formulation of FBX-SNELTs

FBX-SNELTs were successfully prepared with lactose as the most suitable diluent to give the minimum disintegration time and the best dissolution profile when compared with Avicel PH-101 or starch-containing formulations. Fifteen formulations were successfully prepared according to BBD (Table 3) and investigated for their in vitro disintegration (Figure 4a) and in vitro dissolution (Figure 4b–d). Mannitol was added to provide adequate hardness to the SNELTs and improve the stability of the final product as reported in a previous study of lyophilized cyclophosphamide [42]. Gelatin, as a water-soluble polymer, served as a matrix-forming binder to maintain mechanical strength during manufacturing and patient handling [43]. Fumed silica, which is an insoluble excipient with a large surface area, was used to increase the amount of adsorbed drug on its surface and facilitate its distribution in the buccal cavity upon disintegration, and consequently enhance its absorption via the mucosal membrane [44]. Xylitol was used to enhance the hardness of the SNELTs and to impart a desirable sweet taste as reported in a previous study on taste masking of the oral administration of ranitidine disintegrating tablets [45]. Croscarmellose sodium, lactose, and xylitol were subsequently added as superdisintegrant, diluent, and sweetening agents, respectively. All the components were homogeneously mixed until the formation of a slurry that was poured into empty pockets of tablet blister packs that were finally lyophilized. Furthermore, HPMC and gelatin solution were successfully used in previous work in the preparation of SNELTs [23,46].

### 3.6. Evaluation of the Prepared FBX-SNELTs

The prepared FBX-SNELTs were evaluated for various parameters. The weight uniformity of SNELTs for all the formulations met the pharmacopeia requirement, and the results ranged between 127.2 ± 1.19 and 135.3 ± 0.7 mg, suggesting the proper formulation of the SNELTs. The thickness of the SNELTs ranged from 5.014 ± 0.19 to 5.170 ± 0.45 mm. The FBX content was found to be more than 93% for all formulations, within the pharmacopeia limits of 90.0% to 110.0%. The friability test for all formulations was less than 1%, which complies with the pharmacopeia specification, and indicates good mechanical strength. These results indicate the ability of the SNELTs to resist mechanical stress conditions during handling.

### 3.7. In Vitro Disintegration Study

The in vitro disintegration time for the formulations ranged from 1.33 min (SNELTs-3) to 5.83 min (SNELTs-9) as demonstrated in Figure 4a. There is a direct relationship between the disintegration time and the percentage of hydroxypropyl methylcellulose (X_3_), and the concentration of gelatin solution (X_2_) (Figure 5a). On the other hand, the disintegration time was inversely proportional to the percentage of croscarmellose sodium (X_1_) (Figure 5a). Formulations containing the highest level of either X_2_ or X_3_ and the lowest level of X_1_ showed the longest disintegration time, as shown in formulations SNELTs-9 and SNELTs-15 (Figure 4a). In contrast, formulations containing the lowest level of either X_2_ or X_3_ and the highest level of X_1_ showed the shortest disintegration time, as in formulations SNELTs-3 and SNELTs-12 (Figure 4a), which is in agreement with reported results [47]. For instance, Marais et al. reported that the increase in croscarmellose sodium concentration reduced the disintegration time of furosemide tablets. This finding may be due to its ability to enhance the rate and extent of liquid uptake and penetration into the tablets, hastening their breakdown and disintegration [47]. AlHusban et al. noticed that the disintegration time of clonidine tablets was decreased by decreasing the concentration of gelatin in the solution [48]. Also, Dave et al. reported a similar finding for chlorpheniramine maleate lyophilized tablets. The formulation containing a high concentration of gelatin showed a longer disintegration time, which the investigators attributed to slow uptake of water from the medium and an increase in the swelling disintegration time [22]. Also, the presence of HPMC (X_1_) as a matrix-forming polymer with high concentration increased the disintegration time of the tablet, which could be attributed to the formation of a high level of cross-linking polymer network that decreases the tablet porosity and increases its hardness [49].

### 3.8. In Vitro Dissolution Study

The release profiles for the fifteen formulations are presented in Figure 4b–d. The cumulative percent of FBX from all formulations ranged from 51.8% (SNELTs-15) to 99.2% (SNELTs-12). The results illustrate that there is a relationship between disintegration time and the dissolution profile. The formulations with the shortest disintegration time, SNELTs-3, SNELTs-4, and SNELTs-12, showed the highest cumulative released amount of FBX. In contrast, the formulations with the longest disintegration time, SNELTs-5, SNELTs-9, and SNELTs-15, showed the lowest cumulative released amount of FBX. This result, which is attributed to the short disintegration time of SNELTs, led to the rapid breakdown of the tablet into small particles that increased the surface area exposed to the medium and consequently enhanced the dissolution of the drug [50].

### 3.9. Response Surface Methodology for Optimization of FBX-SNELTs

BBD was utilized for the optimization of FBX-SNELTs to minimize disintegration time and maximize FBX dissolution within 1 h. The experimental design matrix with different levels of the factors is compiled in Table 3.

#### 3.9.1. Influence of the Independent Variables on Tablet Disintegration (Y_1_)

For the fast disintegration of tablets, it is essential to ensure that tablets break down rapidly into smaller fragments to yield the largest possible surface area available to the dissolution media [51]. The prepared FBX-SNELTs showed marked variation in disintegration times ranging from 1.33 min for formulation (SNELTs-3) to 5.83 min for formulation (SNELTs-9). The effect of the investigated factors on the disintegration timeis demonstrated in the 3D response surface plots (Figure 5b–d). A polynomial equation (4) was generated as follows:In vitro disintegration time (Y_1_, min) = 498.75 − 98.75 X_1_ − 26.667 X_2_ + 313.333 X_3_ + 6.354 X_1_^2^ + 1.25 X_1_ X_2_ − 55.0 X_1_ X_3_ + 5.417 X_2_^2^ + 30.0 X_2_ X_3_ + 86.667 X_3_^2^(4)

The ANOVA results (Table 4) and Pareto chart (Figure 5a) depict a significant negative effect of X_1_ and the interaction term (X_1_X_3_) on the disintegration time of FBX-SNELTs (Y_1_), with *p*-values of 0.0001 and 0.0002, respectively. This result shows the presence of an inverse relationship between these factors and the disintegration of tablets. As the concentration of croscarmellose increased, the tablet disintegrated faster. However, ANOVA also showed a significant positive effect of X_2_, X_3,_ and the quadratic term of X_1_ and the interaction term (X_2_ X_3_) corresponding to the disintegration time of FBX-SNELTs with *p*-values of 0.0004, 0.0001, 0.0004 and 0.0484, respectively. As a result, formulations containing the lowest level of either X_2_ or X_3_ and the highest level of X_1_ showed the shortest disintegration times. This result could be due to the high porosity of tablets and the adequate amount of superdisintegrant required to swell and break down the tablets as previously reported by Elkordy et al. [51]. Croscarmellose sodium is a hydrophilic polymer and absorbs many times its weight in water to rapidly swell the tablets. This wicking action will spontaneously replace the tablet–air interface with a tablet–water interface and maintain a capillary flow leading to rapid disintegration [52].

On the other hand, the formulations containing a high level of either X_2_ or X_3_ and a low level of X_1_ showed the longest disintegration time. Gelatin and HPMC were used as structure-forming excipients in these formulations. The disintegration of the tablet is affected by pore structure and bonding structure within the tablet as the pores facilitate rapid water penetration into the tablet to rupture the bonds and break down the tablets into small fragments. The binder effect of gelatin and HPMC in these formulations decreases the porosity of tablets, and thus the medium penetration into the tablets decreases, which slows down the disintegration process [53]. Liew and Peh reported that HPMC prolonged the disintegration time of tablets due to the formation of a high level of cross-linking polymer network that decreases the porosity of the tablets [49]. Also, increasing the concentration of gelatin increases bond strength between the tablet particles leading to an increase in tablet hardness and disintegration time as previously reported by Widjaja et al. [54].

#### 3.9.2. Influence of Independent Variables on Cumulative FBX Release (Y_2_)

The dissolution profiles of FBX-SNELTs formulation are represented in Figure 4b–d. The cumulative FBX release from the SNELTs showed marked variation, ranging from 51.8% (SNELTs-15) to 99.2% (SNELTs-12). A polynomial equation (5) was generated as follows:Cumulative FBX release (Y_2_, %) = 30.438 + 13.631 X_1_ + 6.579 X_2_ − 89.633 X_3_ − 0.354 X_1_^2^ − 0.225 X_1_X_2_ + 0.05 X_1_X_3_ − 1.417 X_2_^2^ − 2.7 X_2_X_3_ + 77.733 X_3_^2^(5)

Statistical analysis (Table 4) showed a significant positive effect of X_1_ and X_3_^2^ on the dissolution of FBX from SNELTs, with *p*-values of 0.0001 and 0.0226, respectively. This revealed a direct relationship between X_1_ and Y_2_, i.e., percentage increase of croscarmellose directly correlates with a cumulative increase in FBX release. On the other hand, X_3_ showed a significant negative effect on the cumulative FBX release from SNELTs as demonstrated in the Pareto charts in Figure 6a. The effects of the studied factors on cumulative FBX release are graphically illustrated in the 3D response surface plots shown in Figure 6b–d. It is evident that the dissolution profile of FBX-SNELTs formulations containing the lowest level of X_3_ and the highest level of X_1_ showed the highest cumulative release of FBX from SNELTs. On the other hand, the lowest cumulative release of FBX was observed in the formulation containing the highest level of X_3_ and the lowest level of X_1_. Drug dissolution is highly dependent on the tablet disintegration; as a result, the higher the concentration of superdisintegrant used, the higher the cumulative drug released, as previously reported by Tanuwijaya et al. [55]. This result could be attributed to the high amount of superdisintegrant that causes the rapid disintegration and breakdown of SNELTs into small particles, increasing the surface area that is exposed to the medium and enhancing its dissolution, and vice versa [56].

#### 3.9.3. Optimization

Numerical optimization following the desirability function approach was applied to predict the optimum FBX-SNELT composition with minimum disintegration time and maximum FBX dissolution within 1 h. This optimized FBX-SNELT contains 20 mg FBX dissolved in 100 mg SNEDS, 10 mg of both fumed silica and lactose, and 5 mg of both mannitol and Xylitol. These ingredients were dispersed in gelatin solution and mixed with the optimum concentrations of both HPMC and croscarmellose sodium. The optimum level of the independent factor was found to be 5.73% of croscarmellose, 1.93% of gelatin solution, and 0.43% of HPMC. The optimized formulation achieved the desirability requirements with a function of 0.829. The prepared optimized FBX-SNELTs complied with the pharmacopeia requirement and specifications for weight uniformity, thickness, content uniformity, and friability. Also, the optimized FBX-SNELTs exhibited a disintegration time of 2.74 min with a 75.5% cumulative release of FBX. The observed parameters are in good agreement with the predictions, with a percentage error of less than 5%.

The cumulative release of FBX from the optimized FBX-SNELTs was compared to the marketed FBX tablets. Figure 7a shows that the release of FBX from SNELTs was significantly higher and faster than from the marketed FBX tablet, the former showing a release of ~75% in 1 h compared to ~40% by the marketed FBX, also in 1 h. Interestingly, FBX-SNELTs released the same amount of FBX in 10 min compared to the release by the marketed FBX tablet in 1 h. This enhancement of FBX dissolution appears to improve its absorption and hence its oral bioavailability as reported [23,57].

### 3.10. In Vivo Pharmacokinetic Study in Healthy Human Volunteers

The plasma concentration–time profiles of FBX after the oral administration of a single dose of the optimized FBX-SNELTs and the marketed FBX tablets are compared in Figure 7b. The pharmacokinetic parameters of the clinical study are represented in Table 5. The results indicate that the maximum plasma concentration (C_max_) of FBX-SNELTs was 1340.0 ± 134.0 ng/mL within 45 min (T_max_), compared to the marketed FBX tablets for which it was 773.5 ± 117.6 ng/mL within 120 min (T_max_). These findings meant that SNELTs improved the rate and extent of FBX absorption. Also, FBX-SNELTs showed a higher area under the curve (8885.9 ± 1578.3 ng/mL·h) in comparison to the marketed tablets (6069.9 ± 1640 ng/mL·h). The relative bioavailability of FBX in the SNELT formulation was 146.4% compared to the marketed tablets. The obtained results suggest that the incorporation of FBX in SNEDS with minimum globule size, loading onto a carrier with large surface area, and lyophilizing in fluffy porous tablets lead to an increase in the rate and extent of absorption as well as improving the oral bioavailability of the drug [23]. The improved absorption of FBX was probably due to the enhanced solubilization of FBX that could be directly absorbed without the dissolution step, which is considered the rate-limiting step for drug absorption in Biopharmaceutical Classification System (BCS) Class II compounds [57].

## 4. Conclusions

The study establishes FBX-SNELTs to improve the poor bioavailability of FBX, and importantly to enhance its clinical usage. The optimized FBX-NE is composed of 10% castor oil, 40% PEG 40 stearate, and 50% Transcutol and led to a SNEDS with 175.7 nm globule size and 91% stability index. A Box–Behnken design was utilized to optimize the level of the lyophilized tablet excipients to develop FBX-SNELTs with minimum disintegration time and maximum drug release. The optimized FBX-SNELTs are composed of 5.73% croscarmellose sodium, 1.93% gelatin solution concentration, and 0.43% hydroxypropyl methylcellulose. This formulation exhibited a disintegration time of 2.74 min and released 75.5% of the FBX within 1h. Comparing the relative bioavailability and the pharmacokinetics parameters of optimized FBX-SNELTs with marketed FBX tablets in healthy human volunteers showed a significant improvement in FBX bioavailability in the developed SNELTs (146.4%). The results from these in vitro and in vivo studies support the usage of FBX-SNELTs in the treatment and management of gout patients due to their superiority in enhancing FBX bioavailability.

## Figures and Tables

**Figure 1 pharmaceutics-12-00534-f001:**
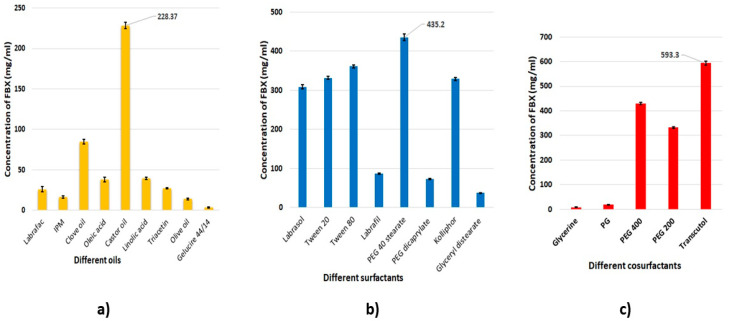
Solubility of FBX in different (**a**) oils, (**b**) surfactants, and (**c**) cosurfactants.

**Figure 2 pharmaceutics-12-00534-f002:**
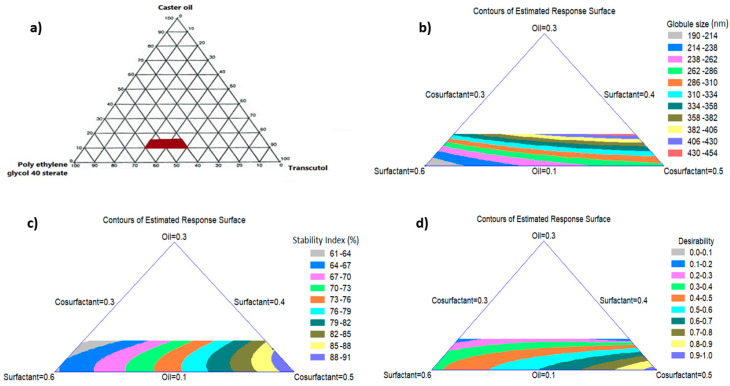
Pseudo-ternary phase diagrams of the selected nanoemulsion system (**a**), Two-dimensional (2D) contour plots of the estimated response surface for the effect of variables on FBX-SNEDS (**b**–**d**). The red area in figure (**a**) represents the clear nanoemulsion region that was selected as a border of the mixture experimental design space.

**Figure 3 pharmaceutics-12-00534-f003:**
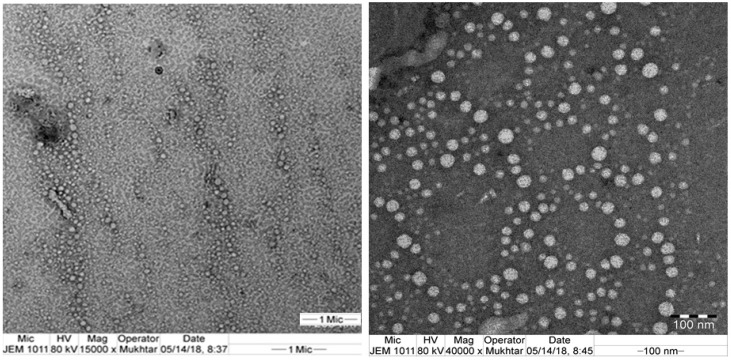
Photomicrographs of the optimized FBX-SNEDS by TEM image.

**Figure 4 pharmaceutics-12-00534-f004:**
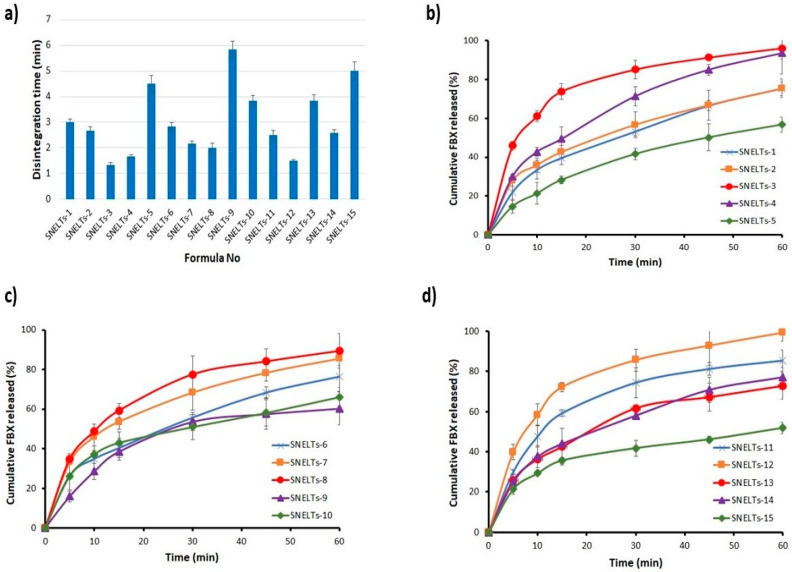
Disintegration (**a**) and dissolution profiles (**b**, **c**, and **d**) of the prepared SNELTs.

**Figure 5 pharmaceutics-12-00534-f005:**
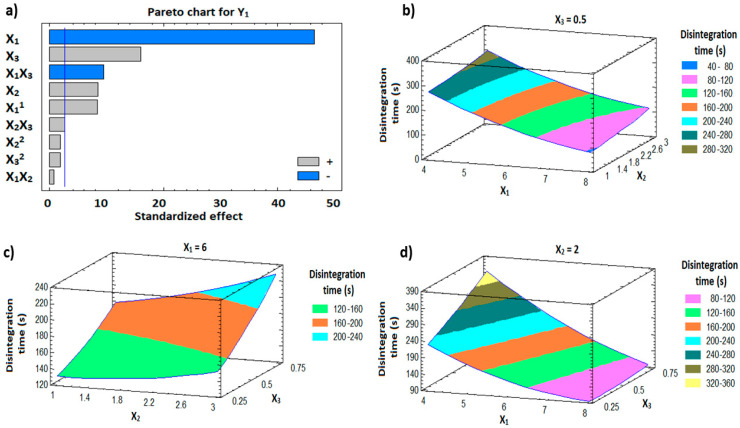
Pareto chart (**a**) and response surface plots (**b**–**d**) showing the effect of the independent variables on the disintegration time (Y_1_).

**Figure 6 pharmaceutics-12-00534-f006:**
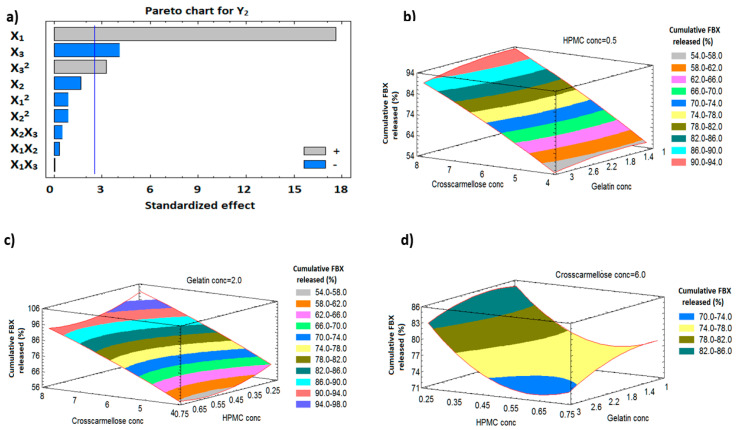
Pareto chart (**a**) and response surface plots (**b**–**d**) showing the effect of the independent variables on cumulative FBX release (Y_2_).

**Figure 7 pharmaceutics-12-00534-f007:**
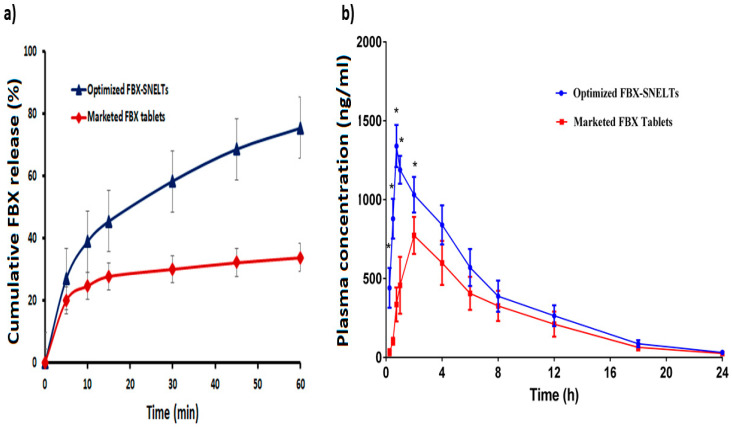
Comparative release profiles (**a**) and plasma concentration–time curves (**b**) between the optimized FBX-SNELTs and marketed tablets.

**Table 1 pharmaceutics-12-00534-t001:** Components of the mixture design and their selected levels.

Component	Level	
	Low	High
Oil percentage (X_1_)	10	15
Surfactant percentage (X_2_)	40	60
Co-surfactant percentage (X_3_)	30	50

**Table 2 pharmaceutics-12-00534-t002:** Composition matrix and the observed mean globule size and stability index of FBX-NE formulations as suggested by the mixture design.

Formula Code	Mixture Components			Dependent Responses	
	X_1_ (%)	X_2_ (%)	X_3_ (%)	Y_1_ (nm)	Y_2_ (%)
**NE-1**	10	60	30	202.2	61
**NE-2**	**10**	**40**	**50**	**175.7**	**91**
**NE-3**	15	55	30	355.7	59
**NE-4**	15	40	45	452.8	85
**NE-5**	11.25	54.375	34.375	210.7	72
**NE-6**	11.25	44.375	44.375	288.7	80
**NE-7**	13.75	51.875	34.375	366.9	70
**NE-8**	13.75	44.375	41.875	389.3	78
**NE-9**	10	50	40	256.3	75
**NE-10**	12.5	57.5	30	232.7	63
**NE-11**	12.5	40	47.5	347.5	89
**NE-12**	15	47.5	37.5	401.3	69
**NE-13**	12.5	48.75	38.75	328.9	73
**NE-14**	10	60	30	197.5	65

**Table 3 pharmaceutics-12-00534-t003:** Independent factor percentages in the formulations of FBX-SNELTs in a randomized order as suggested by a Box–Behnken design.

Formula Code	X_1_	X_2_	X_3_
SNELT-1	6.0	1.0	0.75
SNELT-2	6.0	2.0	0.5
SNELT-3	8.0	1.0	0.5
SNELT-4	8.0	2.0	0.75
SNELT-5	4.0	1.0	0.5
SNELT-6	6.0	2.0	0.5
SNELT-7	6.0	1.0	0.25
SNELT-8	8.0	3.0	0.5
SNELT-9	4.0	2.0	0.75
SNELT-10	4.0	2.0	0.25
SNELT-11	6.0	3.0	0.25
SNELT-12	8.0	2.0	0.25
SNELT-13	6.0	3.0	0.75
SNELT-14	6.0	2.0	0.5
SNELT-15	4.0	3.0	0.5

Abbreviations: X_1_, groscarmellose sodium percentage; X_2_, gelatin solution concentration; X_3_, hydroxypropyl methylcellulose percentage.

**Table 4 pharmaceutics-12-00534-t004:** Statistical analysis of variance (ANOVA) of the responses (Y_1_ and Y_2_) results.

Factors	Disintegration Time (Y_1_), min			Cumulative Release after 60 min (Y_2_), %		
	Estimate	*F*-Ratio	*p*-Value	Estimate	*F*-Ratio	*p*-Value
X_1_	−190.0	2166.00	0.0001 *	35.83	311.63	0.0001 *
X_2_	35.0	73.50	0.0004 *	−3.58	3.10	0.1384
X_3_	65.0	253.50	0.0001 *	−8.5	17.54	0.0086 *
X_1_X_1_	50.83	71.56	0.0004 *	−2.83	0.90	0.3864
X_1_X_2_	5.0	0.75	0.4261	−0.9	0.10	0.7665
X_1_X_3_	−55.0	90.75	0.0002 *	0.05	0.00	0.9868
X_2_X_2_	10.83	3.25	0.1313	−2.83	0.90	0.3864
X_2_X_3_	15.0	6.75	0.0484 *	−1.35	0.22	0.6579
X_3_X_3_	10.83	3.25	0.1313	9.72	10.58	0.0226 *
R^2^	99.81			98.58		
Adj. R^2^	99.48			96.01		
SEE	5.77			2.87		
MAE	2.44			1.23		

Note: * Significant effect of factors on individual responses. Abbreviations: X_1_, croscarmellose sodium percentage; X_2_, gelatin solution concentration; X_3_, Hydroxypropyl methylcellulose percentage; X_1_X_2_, X_1_X_3_, X_2_X_3_, the interaction term between the factors; X_1_X_1_, X_2_X_2_, and X_3_X_3_, the quadratic terms between the factors; R^2^, R-squared; Adj-R^2^, adjusted R-squared; SEE, standard error of estimate; MAE, mean absolute error.

**Table 5 pharmaceutics-12-00534-t005:** Pharmacokinetic parameters of the optimized FBX-SNELTs compared to the marketed FBX tablets (mean ± SD; *n*  =  6).

PK Parameters	Optimized FBX-SNELTs	Marketed FBX Tablets
C_max_ (ng/mL)	1340.0 ± 134.0	773.5 ± 117.6
T_max_ (min)	45.0 ± 0.0	120.0 ± 0.0
t_1/2_ (h)	4.0 ± 0.27	4.28 ± 0.50
AUC_0–t_ (ng/mL h)	8885.9 ± 1578.3	6069.9 ± 1640.0
AUC_0–inf_ (ng/mL h)	9068.6 ± 1590.0	6230.7 ± 1715.7
AUMC_0–inf_ (ng/mL h^2^)	60,175.0 ± 12,212.0	46,481.8 ± 15,071.3
K_el_ (h^−1^)	0.173 ± 0.01	0.165 ± 0.02
MRT (h)	6.61 ± 0.19	7.39 ± 0.39
Relative BA (%)	146.4	-

Abbreviations: C_max_, the maximum plasma concentration; T_max_, Time to maximum plasma concentration; t_1/2_, the elimination half-life; AUC_0–t_, the area under the plasma concentration-time curve from zero time to the last measurable concentration; AUC_0–inf_, the area under the plasma concentration-time curve from zero time to the infinity; AUMC, the area under the first moment curve; Kel, the terminal elimination rate constant; MRT, the mean residence time; BA, the bioavailability.

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
