# Peer review of "Investigating the Potential of Transmucosal Delivery of Febuxostat from Oral Lyophilized Tablets Loaded with a Self-Nanoemulsifying Delivery System"

_pharmaceutics, 2020, doi:10.3390/pharmaceutics12060534_

Round 1

Reviewer 1 Report

The manuscript contains the investigating the potential of transmucosal delivery of febuxostat from oral lyophilized tablets loaded with self-nanoemulsifying delivery system. Thus, it is suitable for publication in the journal "Pharmaceutics ". However, it is thought to be accepted with reject because this manuscript has the following problems.

  1. In section “Abstact”:

(1) The content of the abstract is insufficient to explain the research content. Supplement the content.

  1. In section “1.Introduction”:

(1) Line 65-69, The description of the lyophilized tablet is insufficient. Explain the advantages and disadvantages by adding a reference

  1. In section “2.Materials and methods”:

(1) There is a lack of explanation of all experimental methods. Please explain in detail with reference.

  1. In section “3.Results and discussion”:

(1) There is a problem with the experimental design in Table 2. Specifically, NE-1 and NE-14 are identical. In addition, it is insufficient to evaluate whether there is no deviation value and significant difference in the ratio of each component and the resulting value. Also, it is not reasonable to compare NE-2 prescriptions with the best ones.

(2) Line 261-274, There is a lack of reasonable explanation for the prescription design of FBX-SNELTs. Please explain in more detail. The reason why you chose gelatin and HPMC also needs to be explained.

(3) There is a lack of specificity in the compositional selection and experimental design and evaluation of prescriptions for manufacturing lyophilized tablets. Naturally, the increased use of the components used as binders indicates disintegration delay and low dissolution.

Reviewer 2 Report

The study entitled “Investigating the Potential of Transmucosal Delivery of Febuxostat from Oral Lyophilized Tablets Loaded with Self-Nanoemulsifying Delivery System” is very innovative and valuable from the drug delivery improvement viewpoint. In general, the authors well documented the results and the discussion of model parameters is very interesting. In my opinion the manuscript can be accepted and therefore I have only some technical comments listed below:

Line 102. Please provide the name of spectrophotometer.

Line 165. Please provide the chromatographer’s name and necessary details, such as column, solvents, calibration curve etc.

Lines 348-349. Are the variables correlated with each other? Please provide the correlation matrix. Assuming the parameters are correlated, can the model be simplified? I encourage the authors to consider performing principal component analysis. Perhaps stepwise regression or genetic algorithm would be also helpful.

Reviewer 3 Report

The manuscript "Investigating the Potential of Transmucosal Delivery of Febuxostat from Oral Lyophilized Tablets Loaded with Self-Nanoemulsifying Delivery System" is an interesting article which deals with the development of novel tablets for transmucosal delivery of febuxostat. The manuscript provides fundamental information on the design, characterization and in vivo pharmacokinetic of the proposed system. The topic of the articles is interesting, it is well written and the abstract underlines the promising findings of the study. I recommend publication in the Pharmaceutics after minor revisions.

In the Introduction, lines 73-74, the process to obtain tablets from FBX-SNEDS is not clearly reported and should be clarify.

In the results section, line 362, the composition (mg of each component) of the resulted tablet and the drug amount are not clearly reported.  Please report accordingly.

The captions of figures and tables must permit to understand the reported data without check the text. According to this, the captions should be improved. In addition, in all the manuscript a large amount of abbreviations has been reported and they make difficult to understand the text. In my opinion, only one or two of the main used words should be abbreviated.
